# Ethical and Societal Issues Occasioned by Xenotransplantation

**DOI:** 10.3390/ani10091695

**Published:** 2020-09-19

**Authors:** Bernard E. Rollin

**Affiliations:** Department of Philosophy, Colorado State University, Fort Collins, CO 80523, USA; bernard.rollin@colostate.edu

**Keywords:** genetic engineering, ethical issues in genetic engineering, xenotransplantation, value-free science, societal issues with xenotransplantation

## Abstract

**Simple Summary:**

The dream of transferring bodily organs from animals to humans goes back to antiquity, as articulated in the myth of Daedalus and Icarus. But the genuine possibility of xenotransplantation is largely a creature of the last half-century. That possibility raises a plethora of genuine ethical issues, as well as spurious ones occasioned by societal ignorance of science.

**Abstract:**

There are three sorts of issues associated with genetic engineering and, by implication, with xenotransplantation. These are dangers associated with the technology, animal welfare issues, and the claim that genetic engineering represents a technology that humans should not embark upon. Using the hearts of pigs for humans in need of transplants has been a major issue in xenotransplantation. There are dangers associated with such use, such as immunological rejection of the organ, endogenous viruses infecting the recipients, and issues of privacy. In addition, the issue of fair distribution of organs arises. Animal welfare issues also arise, most notably the living conditions of the donor animals, issues notably present in confinement agriculture. A major issue emerges from animals’ being kept under conditions that fail to meet the needs dictated by the animals’ biological and psychological natures. Xenotransplantation animals will be kept under deprived laboratory conditions that similarly fail to meet the animals’ natures. This is a significant concern for society in general. There are also issues of “bad ethics” arising from scientists’ disavowal of ethical concerns in science. This in turn, coupled with societal ignorance of science, creates a climate for proliferation of religious and other non-rational concerns, such as the claim that xenotransplantation violates God’s will. These spurious concerns can only be ameliorated when public understanding of science improves, and scientific understanding of ethics increases.

## 1. Introduction 

In 1995, I published what I believe to be the first book ever written on the social and ethical issues associated with the genetic engineering of animals, entitled *The Frankenstein Thing: Ethical and Social Issues in the Genetic Engineering of Animals* [1]. I used the iconic image of the Frankenstein monster as the symbol for society’s view of biotechnology. I disambiguated three different aspects of the myth that I called “There are certain things humans were not meant to do”, “Rampaging monsters”, and “The plight of the creature”. Aspect one referred to the widespread belief, primarily for religious reasons, that certain areas should not be studied. Aspect two referred to dangers attendant on human manipulation of species. Aspect three referred to the suffering of the created being. All of these are strikingly present in the issue of using animal organs for xenotransplantation.

While aspect one was of great social prominence, I did not see it as a genuine ethical issue, though much of society surely did. I in fact enunciated what I called “a Gresham’s law for ethics”. Recall that Gresham’s law affirms that “bad money drives good money out of circulation”. For example, after World War I in Germany it took a wheelbarrow full of deutschemarks to buy a loaf of bread, so devalued was the currency. In such a situation, no one paid off major debts in gold, but rather in worthless currency. Similarly, I argued that “bad ethics drives good ethics out of circulation”, since such aspect one claims as “genetic engineering violates God’s will” are far “sexier” than genuine ethical issues. After Dolly the cloned sheep was announced, for example, a Time/Warner survey conducted the next week revealed that 75% of the U.S. public believed that the creation of Dolly violated God’s will.

Aspect two referred to various kinds of danger or harm associated with genetic engineering; for example, environmental dangers or the potential for new pathogens. While certainly counting as ethical issues, they were equally matters of prudence. Aspect three, the effect of genetic engineering on the welfare of the animals engineered, was a truly ethical issue, as it almost invariably pitted the well-being of the animals against human well-being. And, as we know from the rise of industrial agriculture and the history of animal research, human benefit usually wins.

The template I have just outlined works very well for all aspects of biotechnology including agriculture, animal research, creation of animal models of human genetic disease, and the topic at hand, xenotransplantation. In fact, xenotransplantation provides a textbook example of all of the above categories. Approximately 3500 to 4000 people seek heart transplants per year in the U.S. As a result, alternatives to human donors have been sought. One putatively plausible alternative has been to use animals, specifically pigs, although historically baboons and other animals have been utilized. There are no cases reported of humans surviving with pig hearts.

Consider the use of pig hearts for people needing heart transplants. (This is overwhelmingly the primary use of xenotransplantation that has been discussed.) There exist very real *dangers* in using animal organs for transplantation, specifically on microbiological, physiological, and immunological levels. There are also *animal welfare issues* associated with such uses, albeit subtle ones. Finally, there are extremely clear examples of *bad ethics supplanting and eclipsing good ethics* in discussions of xenotransplantation.

## 2. Dangers Associated with Xenotransplantation

Pigs have been the animals of choice for organ donation, specifically heart, but also kidneys and islet cells of the pancreas. There are numerous reasons for this. Pig organs, particularly the heart, are comparable in size to that of humans. In addition, pigs, as domesticated animals, can easily be raised in relatively controlled confinements, which is not the case with nonhuman primates. In addition, because of genetic relatedness with nonhuman primates, there is a greater risk of disease spread in the case of primates. Additionally, pigs have relatively large litters—up to a dozen piglets at a time—and have a relatively short gestation period; three months, three weeks, and three days. For these reasons, certain governments, for example that of the UK, have strongly discouraged and even banned nonhuman primates as candidates for transplantation. See the Nuffield Council on Bioethics’s comprehensive report on xenotransplantation [2]. (The logic implicit in the use of pig hearts for human transplantation also holds in the case of other transplantable organs such as kidneys and livers, and islet cells for elimination of diabetes.)

As of mid-April 2019, an Israeli scientist has announced achieving three-dimensional printing of the human heart [3]. It is as yet too early to tell if this is a practicable alternative to xenotransplantation. Since it is created from the patient’s own cells, the problem of immunological rejection may be circumvented.

The most significant issue with using animals for a source of transplanted organs (xenotransplantation) for humans is immunological rejection of the organ, with the human immune system recognizing the foreign organ as “not-self” and correlatively rejecting it. In what is known as “hyper-acute rejection”, the body begins to reject the organ virtually as soon as it is implanted [4].

In the case of genetically engineered pigs, however, a small amount of human genetic material can be injected into the developing pig embryo, so that the resulting piglet is not recognized as foreign. In 2016, National Institutes of Health (NIH) researchers announced that a pig heart had been kept alive in a baboon for three years. The use of immune-system-suppressing drugs has also reduced the probability of rejection.

An additional major danger with xenotransplantation are the endogenous retroviruses carried by pigs, which are capable of making humans very sick although not harming the pigs [4]. These viruses are called porcine endogenous retroviruses (PERVs). One salient example of a PERV was the virus likely responsible for an epidemic of swine flu in 2009, which took about a quarter of a million human lives. Another example is the Nipah virus, which caused an epidemic of encephalitis in Singapore and Malaysia. One can speculate that some of these viruses could conceivably not be recognized until such time as they created disastrous epidemics. Antibiotic-resistant bacteria are also capable of being transplanted into humans through the use of swine organs. In addition to infecting a grant recipient, some of these diseases may spread “horizontally” across a community, serving as a danger to public health. All of this clearly illustrates the great extent to which xenotransplants represent grave danger. 

In addition, concerns for safety for the transplant recipient, as well as for public health, put a severe limitation on the privacy of such recipients, as they need to be monitored for pathogens dangerous to them and others, for what may possibly be very extended lengths of time, which in turn affects the privacy not only of the recipient, but also of his or her family, friends, work associates and others with whom he or she is in contact. Thus, there arises an ethical tension between the good of the recipient and the good of society. Correlatively, it is extremely unlikely that the identity of those receiving xenotransplants can be kept confidential, so it is quite possible that such people will be seen as significantly different, or even as “freaks”, particularly if the recipient is a child and will likely be subject to taunting with locutions like “pig heart” regularly thrown at them, which in turn can cause significant psychological damage. Another ethical issue associated with xenotransplantation is the question of how such organs would be distributed. What is a fair way to assure equitable distribution of both risks and benefits [5]? 

In sum, there are a variety of dangers associated with xenotransplantation, both for recipients and for the public at large. As yet, however, none of these projected dangers have come to pass, in part because there have been no successful xenotransplants. Governments have been extremely cautious about allowing unrestricted research into xenotransplantation to proceed, presumably because of societal hesitation regarding all aspects of biotechnology.

## 3. Animal Welfare Issues Associated with Xenotransplantation

Some people are instinctively horrified by the idea of raising pigs to put their organs into human bodies. A reality check should diminish that reaction. The number of animals raised for xenotransplantation will probably number in the thousands. In contrast, the number of pigs killed in a year in the world for food is over a billion [6]! If we can cavalierly kill that many pigs in pursuit of bacon, ham, and sausages, we should surely not be horrified at killing them in order to save human lives.

The ethical issue of concern to the majority of society is not the killing of animals, as the above number attests. There is little reason to believe that animals fear death or anticipate that it is coming. Lacking human linguistic capacity, animals cannot conceive of death, described by Heidegger as “awareness of the possibility of the impossibility of [their] being”. While animals certainly communicate, they do not possess the ability to transcend the moment and imagine the future. In all fairness, there are those who do see death as a welfare issue, since opportunities for future pleasures are denied to the animal.

Rather, what troubles society in general is the question of how the animals live, and the unnatural conditions under which they are housed and maintained. Attesting to this point has been the elimination of such severe systems as confinement veal production in wooden crates where the animals are tethered so as to keep the flesh soft, and fed a diet designed to produce borderline anemia so that consumers can eat “white veal”. Gestation crates—essentially 6′ by 2.5′ by 3′ cages that are so small that they do not allow sows to turn around or even stand up—have been banned in Scandinavia for decades, and the EU committed beginning in 2013 to phase them out. The UK and New Zealand have already done so. Smithfield, the largest pork producer in the U.S., began eliminating gestation crates in 2008, and finished the process in 2019.

Were pigs for food raised the way they were traditionally kept, outdoors on pasture with barns to retreat to in severe weather, the severe confinement necessitated by keeping swine used for organ transplants would be blatantly unacceptable. Traditional rearing of all farm animals was accomplished by placing the animals under conditions meeting the needs following from their biological and psychological natures—what Aristotle called their *telos* [7], the “pigness” of the pig, the “cowness” of the cow. In fact, raising animals under these natural conditions was an essential component of the “good husbandry” that was the key to agricultural success for 10,000 years, until after the Industrial Revolution.

“Husbandry” is derived from the Old Norse phrase *hus bond*, meaning “bonded to the household”. Husbandry has been termed “the ancient contract with animals”, where, as in any fair contract, both parties benefit from the relationship. Virtually until the 20th century, husbandry was the key concept in animal agriculture. 

The essence of husbandry was *care*. Humans put animals into the most optimal environment congenial to the animals’ not only surviving but thriving, the environment for which they had evolved and been selected. The better the animals did, the better farmers did. Sanctioned by their own self-interest, humans provided farm animals with sustenance, shelter, protection from predation, such medical attention as was available, help in birthing, food during famine, water during drought, safe surroundings, and comfortable appointments. The Noah story is emblematic of this “ancient contract”. In the Noah story, we learn that even as God preserves humans, humans preserve animals. In the 23rd Psalm, the Psalmist points out that God is to humans as shepherd are to sheep [8].

The singular beauty of husbandry is that it was at once an ethical and prudential doctrine. It was prudential in that failure to observe husbandry inexorably led to ruination of the person keeping animals. Not feeding, not watering, not protecting from predators, not respecting the animals’ physical, biological, physiological, and psychological needs and natures, meant your animals did not survive and thrive, and thus neither did you. Thus, no formally articulated animal ethic was needed. Animal husbandry in essence became the basis for what was the newly civilized society and the leisure time necessitated by the development of culture, as Thomas Hobbes pointed out in *Leviathan*. Thus, husbandry was about putting square pegs into square holes, round pegs into round holes, and creating as little friction as possible while doing so. The industrialization of confinement agriculture, as it developed in the 20th century, enabled the forcing of square pegs into round holes, round pegs into square holes, i.e., animals are placed into confinement systems for which they are not naturally suited, by virtue of the development of what I have called “technological sanders”, such as antibiotics, vaccines, air-handling systems, that severed the traditional connection between productivity and welfare. The use of antibiotics for disease prevention and growth promotion has become a major societal issue, given the risk of developing antibiotic-resistant pathogens [9].

Swine raised for organ transplantation will certainly not be kept under traditional husbandry conditions. Rather, they will be kept much in the manner of laboratory animals, under confined, sterile conditions that minimize the risk of pathogen proliferation and keep the animals sufficiently healthy to provide a (relatively) safe source for transplantation. Although such conditions are far better than agricultural conditions in terms of animal health, they are equally deficient in accommodating the animals’ biological and psychological natures. Thus, we may confidently affirm that while such animals will be healthier, and thus have increased welfare in a single dimension, their overall welfare, i.e., the respect for the needs emerging from their *telos*, will be no better [10].

There are numerous accounts of “animal welfare” circulating in contemporary thought, ranging from an animal experiencing more pleasure than pain (the utilitarian view) to my own account based in the thinking of Aristotle. Aristotle believed that things in the natural world would best be explained *teleologically*, i.e., by reference to their basic function or nature, on the model of biological organisms. This is what Aristotle [7] called their *telos* (i.e., their *nature*). Post-Newtonian science is mechanistic, with physics serving as the model for good explanations. When I began to work on animal ethics, it dawned on me that *telos* was the most reasonable vehicle for describing our moral obligations to animals. *Telos* accommodates an animal’s biological and psychological needs and nature—the “pigness of a pig”, the “dogness of a dog”. As such, it is extremely congenial to common sense. Implicit reference to *telos* is what militates against the acceptability of small cages for pigs or veal calves, or any cages for zoo animals. Furthermore, the notion of *telos* fits beautifully with Western political theories according rights to humans based on the protection of their human nature, and accords beautifully with the Platonic dictum that ethics should proceed from pre-existing ethics.,(i.e., that animal ethics would be best derived from our established human ethics).

When I discuss animal welfare with the scientific community, I am usually told that it is solely a matter of “sound science”. While science is certainly relevant, questions of animal welfare are at least partly “ought questions”, questions of ethical obligation. The concept of animal welfare is in part an ethical concept to which science brings relevant data. When we ask about an animal’s welfare, or about a person’s welfare, we are asking about *what we owe the animal and to what extent.*

In the early 1980s, the U.S. agricultural community issued a document known as the CAST Report, which affirmed that “what we owe animals and to what extent is simply what it takes to get them to create profit” [11]. This implies that animals are well off if they only have food, water, and shelter. At about the same time, the British Farm Animal Welfare Council (FAWC) declared that animal welfare includes the animal’s physical and mental state and implies both “fitness and a sense of well-being” [12]. This was in turn defined in terms of the famous five freedoms: freedom from hunger and thirst, freedom from discomfort, freedom from pain injury or disease, freedom to express normal behavior, and freedom from fear and distress. (The FAWC definition is closely connected to what I call *telos.*)

Clearly the two definitions contain very different notions of our moral obligations to animals (and there is an indefinite number of other definitions). Which is correct cannot be decided by gathering facts or doing experiments—indeed, which ethical framework one adopts will in fact determine the shape of science studying animal welfare. Thus, sound science does not determine your concept of welfare; rather your concept of welfare determines what counts as sound science! Certainly, society leans in the direction of the FAWC definition. In my books on animal ethics [10,13], I have argued that respect for animal nature (*telos*) is the fundamental basis for the moral status of animals, even more so than the ability to feel pain. (Some animals caught in steel-jawed traps will chew their legs off to escape!).

One can surmise that a *telos-*based/FAWC view of welfare will continue to dominate societal pressure for farm animal welfare. However, in the case of swine organs being used to save lives, the situation is not so clear. In the case of dogs and primates used in science, society has moved relentlessly in the direction of environments for these animals that at least partially accommodate their social, biological, and psychological natures. Undoubtedly, conditions for keeping swine destined for organ donation will continue to improve, although within the constraints of *gnotobiotics*, i.e., control of pathogenic microorganisms. There is no reason that such swine cannot enjoy significantly enriched confinement as societal awareness of farm animal welfare continues to grow. In the end, as I have said many times before in many contexts, they are having their lives taken by humans; the least we can do for them is to make sure that what life they do have is a good and pleasant one. This should include adequate space, an enriched environment, preservation of social groups, good handling by stockpeople, adequate veterinary care, proper diet, climate control. However, the attempt to keep these animals free of pathogens significantly limits their environment and thus their ability to actualize their *telos*, i.e., their psychological and biological natures. It is unlikely, given these concerns, that animals could be kept outdoors in a pastoral manner, as controlling the extensive environment microbiologically is near impossible.

## 4. Issues of “Bad Ethics”

Many articles on xenotransplantation stress the importance of societal involvement in the formation of policy regarding xenotransplantation, for example [14]. The ethical issues we have discussed so far are genuinely ethical issues or issues of danger. Thus, it seems to be obligatory to also pay attention to what we have referred to as “bad ethics”. We shall now demonstrate that integrally involved in bad ethics is appalling social lack of understanding of science, as well as the scientific community’s denial of the relevance of ethics to science.

As I have described before, there is a perfect storm combining these two factors that militate in favor of the proliferation of bad ethics [15]. Using the United States as my source, we can get a clear sense of how ignorant the general public is about science. In a 2004 paper, neuroscientist and physician Dr. Keith Black affirmed that research shows the following: “One half of the American public does not know the earth goes around the sun once a year. A 1996 National Assessment of Educational Progress survey found that 43 percent of high school seniors did not meet the basic standard for scientific knowledge” [16]. A 2015 Pew Charitable Trusts report on scientific literacy indicates that 40% of the U.S. public believes that humans coexisted with the dinosaurs [17]. And a 2015 survey showed that 80% of the public wanted mandatory labeling of food indicating the presence of DNA [18]!

A 2014 Gallup poll cited in a National Center for Science Education report stated that only a very small percentage (19%) of the U.S. public has accepted the naturalistic position that “man has developed over millions of years from less advanced forms of life. God had no part in this process” [19]. Correlatively, a 2017 Gallup poll indicated that “the percentage of Americans who actively want creation not evolution to be taught in schools may seem relatively modest at 30%, but this number is significantly larger than the percentage who actively want evolution to be taught over creation” [20].

Things are no better from the science perspective. Science has done little to address societal ignorance. Some years ago, I, a philosophy professor with a reasonable background in science, began to plan to teach a full-year introductory honors biology course along with a biologist with interest in philosophy. With all the major changes taking place in biology, particularly in the area of biotechnology, as well as the ever-increasing societal fermentation regarding the use of animals in research, we were concerned that biology students were ill-prepared to engage these concerns. We received funding from the National Science Foundation (NSF) and began looking for suitable textbooks. We were appalled at what we found. A recurrent theme invariably found in the “throat-clearing” introductions to these textbooks (generally making brief mention of Linnaeus, Darwin, etc.) was the claim that science was “value-free” and, more importantly, that science has nothing to do with ethics. Such a position represented a legacy of the philosophical position known as “logical positivism” that dominated science for much of the 20th century.

If one asks most working scientists what separates science from religion, speculative metaphysics, or shamanistic world views, they would unhesitatingly reply that it is an emphasis on validating all claims through sense experience, observation, or experimental manipulation. This component of scientific ideology can be traced directly back to Newton who proclaimed that he did not “feign hypotheses” (*hypotheses non fingo*) but operated directly from experience [21].

The insistence on experience as the bedrock for science continues from Newton to the 20th century, where it reaches its most philosophical articulation in the reductive movement known as Logical Positivism, a movement that was designed to excise the unverifiable from science and, in some of its forms, to axiomatize science so that its derivation from observations was transparent. Examples of positivist targets were Bergson’s (and other biologists’) talk of life force (*élan vital*) as separating the living from the non-living, or the embryologist Driesch’s postulation of “entelechies” to explain regeneration in starfish.

Although logical positivism took many subtly different and variegated forms, the message, as received by working scientists, and passed on to students (including myself), was that proper science ought not allow unverifiable statements, including ethical judgments.

What does all this have to do with ethics? Quite a bit, it turns out. The philosopher Ludwig Wittgenstein, who greatly influenced the logical positivists, once remarked that, if you take an inventory of all the facts in the universe, you will not find it a fact that killing is wrong [22]. In other words, ethics is not part of the furniture of the scientific universe. You cannot, in principle, test the proposition that “killing is wrong”. It can neither be verified nor falsified. So, in Wittgenstein’s view, empirically and scientifically, ethical judgments are meaningless. From this, it was concluded that ethics is outside of the scope of science, as are all judgments regarding values, rather than facts. The slogan that I had in fact learned in my science courses in the 1960s, and which has persisted to the present, and is still being taught in too many places, is that “science is value-free” in general, and “ethics-free” in particular.

It is therefore not surprising that when scientists were drawn into social discussions of ethical issues, they were every bit as emotional as their untutored opponents. It is because their ideology dictates that these issues *are nothing but emotional*, that the notion of rational ethics is an oxymoron, and that he who generates the most effective emotional response “wins”. So, for example, in the 1970s and 1980s debate on the morality of animal research, scientists either totally ignored the issue, or countered criticisms with emotional appeals to the health of children. For example, in one film entitled “Will I Be All Right, Doctor?” (the question asked by a frightened child of a pediatrician), made by defenders of unrestricted research, the response was as follows: “Yes, if *they* leave us alone to do what we want with animals.” So appallingly and unabashedly emotional and mawkish was the film that, when it was premiered at the American Association for Laboratory Animal Science (AALAS) meetings at a subsection of laboratory animal veterinarians, a putatively sympathetic audience, the only comment forthcoming from the audience came from a veterinarian, who affirmed that he was “ashamed to be associated with a film that is pitched lower than the worst anti-vivisectionist clap-trap!”.

Positivist thinkers felt compelled to explain why intelligent people continued to make moral judgments and continued to argue about them. They explained the former by saying that when people make assertions such as “killing is wrong”, which seem to be statements about reality, they are in fact describing nothing. Rather, they are “emoting”, expressing their own revulsion at killing. “Killing is wrong” really expresses “Killing, yuk!” rather than describing some state of affairs. And when we seem to debate about killing, we are not really arguing ethics (which one cannot do any more than you and I can debate whether we like or do not like pepperoni), but rather disputing each other’s facts. So, a debate over the alleged morality of capital punishment is my expressing revulsion at capital punishment while you express approval, and any debate we can engender is over such factual questions as whether or not capital punishment serves as a deterrent against murder.

Perhaps the most extreme example of the denial of ethics occurred when the director of the NIH (the major U.S. government funding agency for biomedical science) made a personal visit to Michigan State University in the early 1990s. Let us bear in mind that there are two presuppositional requirements for being director of the NIH. One is being an outstanding scientist, which this gentleman was. But equally important is a sense of diplomacy allowing one to negotiate very treacherous waters. The students became aware of his visit, and invited him to address the premedical club. A number of students asked him about the ethical issues occasioned by biotechnology. He was apparently unguarded in his remarks, not realizing that a student reporter for the school paper was present. He opined that “though scientific advances like genetic engineering are always controversial, science should never be hampered by ethical considerations” [23]. When I quote that remark to my students, even freshmen, and ask them who said that in the 20th century, they invariably say “Hitler”, and are extremely shocked when I tell them that it was the director of the NIH!

The position of the scientific community on ethics had the predictable effect of encouraging emotionally based religious and non-rational positions articulated by society under the rubric of ethics, as we indicated earlier with regard to the cloning of Dolly the sheep. Thus, science’s denial of ethics when combined with society’s unclear understanding of ethics creates “a perfect storm” blocking a reasonable articulation of the ethical issues of xenotransplantation, and opening the doorway for bad ethics.

Scholars have been carefully tracking European opinions on numerous changes in scientific, social, and cultural life by way of a vehicle known as the Eurobarometer. These surveys have been conducted under the aegis of a sociologist named George Gaskell based at the London School of Economics [24]. Gaskell has a significant interest in biotechnology and in attitudes thereto. In their writings, Gaskell and his associates have checked on attitudes towards six modalities of biotechnology—genetic testing for genetic diseases, cloning human cells and tissues, genetically modified enzymes for producing environmentally friendly soaps, genetically modified crops, genetically modified foods, and xenotransplantation.

Surprisingly, of all these instances of biotechnology, the one receiving the least support was xenotransplantation, by quite a significant majority. This is true each time polls were conducted. For example, in a 1999 article in *Science*, George Gaskell and his associates showed that a higher percentage of the European public opposed xenotransplantation than any other biotechnological modality. Prima facie, this appears to be incomprehensible. After all, xenotransplantation is most directly relevant to saving lives. Gaskell has also repeatedly made the point that moral concerns are more important to the public in evaluating biotechnology than considerations of risk and safety [25]. Obviously, xenotransplantation is seen as presenting weighty moral concerns.

Many people articulated what we have called “spurious moral concerns” (bad ethics), i.e., reasons for rejecting transplanting organs from animals, including concerns about the “unnatural” dimension of such a procedure. Others saw xenotransplantation as “playing God” or “crossing species barriers”. Such claims are easily defeated. If transplanting organs from animals to humans is unnatural, so too are building flying machines, damming the Nile, exterminating pathogenic insects, and the myriad activities in which we engage to make nature more congenial to human life and flourishing. In the same manner, we can put paid to claims that “crossing species barriers violates God’s will”. One can plausibly claim that if God did not wish humans to blur the lines between species, as we have in fact done repeatedly in plant agriculture, why would He have given us the ability to do so? (In fact, many theologians who support biotechnology make precisely this claim.) The key point is that theologically-based opinions do not in and of themselves constitute rational ethical concerns.

I personally believe that much of the opposition to xenotransplantation stems from the appalling degree of social illiteracy regarding transplant science. Many people in society, for example, still retain the ancient belief that the heart is where personality resides and is the locus and seat of emotions, thought, and will, a belief entertained by both the ancient Greeks and the ancient Egyptians. This idea of course persists in the belief that love is founded in the heart, and in such phrases as “he has a kind heart”. (Hence the myth of Cupid firing arrows into the heart.) This naïve view in turn manifests itself in the belief that if one acquires a heart from a pig, one will have the emotions and feelings of a pig, and thus be rendered considerably less than human. Such a naïve view can and does actually impede the development of therapies that fill the perennial need for heart transplants. Essentially, this belief suggests that if one acquires a heart from a pig, one will be transformed into a being possessing the mind and feelings of a pig. Once again, we see the extent to which ignorance of science creates bad ethics, even when what is at stake is preserving life!

To recapitulate, transplanting organs from animals to people is largely condemned because of people’s extremely primitive views of biology and the mind. Once again, we find that bad biology ramifies in bad ethics, which in turn provides a basis for rejecting life-saving biotechnology. Thus, we see that creating a scientifically literate public is essential for societal acceptance of biotechnology. Let us recall that no less an ethical luminary than the Pope condemned the cloning of Dolly the sheep, based upon the belief that “all animals are entitled to a normal birth”. It seems that the Pontiff was unaware of the fact that Dolly had in fact experienced a normal birth! This kind of thinking, or rather lack thereof, serves no one well.

## 5. Conclusions

As we indicated at the beginning of our discussion, and as the rise of xenotransplantation makes plain, biotechnology presents the potential for changing human life in ways that are unprecedented. And, as Gaskell’s research indicates, moral considerations regarding biotechnology are putatively of greater concern to society than even issues of safety. At the same time, societal ignorance of both science and ethics militate strongly against rational solutions to ethical issues, and even elevate non-issues to the highest rank of ethical concerns, for example, “violating God’s will” or operating “against nature”. Such a mistake, as our discussion indicates, can impede using biotechnology to save human life and alleviate suffering.

This in turn indicates the desperate need for better education of both the public and the scientific community in ethics, and of society in general in science, given the abysmal social ignorance of science we illustrated at the beginning of our discussion.

While recognition of this issue is comparatively easy, its resolution is far more difficult. I have both taught and advocated for the integration of ethical issues into science education as presuppositional to rational solutions of ethical issues arising out of scientific developments. I have also seen that such education makes for better scientists who have a sense of social responsibility. I have also argued for the teaching of science, and in particular biological science, to include discussion and elucidation of ethical issues. This has proven to be far more difficult. Impediments to such an approach include the fact that it is historically unprecedented and that most scientists have been brainwashed to believe that science is “value-free in general, and ethics-free in particular”, as demonstrated in our discussion. Further impediments arise out of the fact that most people teaching science have themselves not been educated in ethical issues arising out of science, or even how to begin to approach such issues. For these reasons, it is difficult to know where and how to begin with inculcation of an ethics component into science education. The same problem is evident in the research community’s cavalier attitude towards the ethical issues attendant upon animal use in research, an attitude that distances the scientific community from the general public, as societal concern with the treatment of animals increases exponentially. If, as leaders of the scientific community have repeatedly affirmed, scientific progress is absolutely dependent upon animal use, it surely behooves that community to address societal ethical concerns attendant upon animal use.

I have also indicated in our discussion that the concept of animal *telos* is presuppositional to the development of animal ethics as I have formulated it, and society seems to agree. The question then obviously arises as to whether genetic modification, which can readily be seen as effecting both major and minor changes in *telos*, is morally acceptable. This is certainly not the place for a full discussion of this vexatious issue. But I have argued that as long as *telos* changes do not negatively affect the animals’ lives, i.e., that genetically modified animals are no worse off than their unmodified predecessors, and ideally are better off, there is nothing morally wrong with creating such genetic modifications. In other words, one foundational issue in genetically engineering animals is providing assurance that genetic modification occasions no harm, and ideally, as is the case in genetic modification to prevent disease or ameliorate genetic defects, the animals will be better off in virtue of such modification.

Such considerations will not be addressed until such time as the thinking of scientists is radically changed in order to assure that ethics enters into standard scientific theory and practice, and this can only be accomplished by revolutionary modification of science education.

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
