# Peer review of "Ethical and Societal Issues Occasioned by Xenotransplantation"

_animals, 2020, doi:10.3390/ani10091695_

Round 1

Reviewer 1 Report

General

This paper deals with a very important ethical issue. I enjoyed reading it. It contains very few references and is largely based on the authors’ views, I think. In the case of this author, that is not a problem. However, given its rather quirky style and author-focused referencing, this is a ‘commentary’, not a ‘review’ and certainly not an ‘article’.

The layout does not comply with journal requirements. It does not have a ‘simple summary’ and seems not to have an introduction. None of these are important considerations.

Should there not be a clear statement somewhere that use of xenotransplantation is clearly consistent with a human-centric utilitarian view, but is entirely speciesist?

Specific

L68 Is the effect of the ‘infusion’ of small amounts of human genetic material to change the immunogenicity as a result of the expression of human, rather than pig, immunogenic proteins? If so, this should be said and the work referred to should be referenced. This is an important bit of science.

L83 et seq. These are interesting bits of speculation, and not much more than that. They should be qualified as such.

L104 reference needed for the number.

L106 why does the lack of human linguistic capacity translate to an inability to conceive of death?

L110 ‘probably’?

L115 gestation crates for pigs? They are no longer allowed in the UK or New Zealand and, I think, several other countries, and the restriction in the EU is partial. The author needs to be precise here.

L175 the major problem with the telos approach (which is excellent, of course) is it is very unlikely for a human being to ever have a complete understanding of it, from the animal’s point of view.

L215 Can’t reference like that!

L221 There is a problem using the US as the source. The tsunami of anti-science being unleashed in that nation may possibly be continued and exacerbated after November 2020, with devastating consequences for its society. I do not believe the same can be said of the situation in Europe and most other developed countries, and that needs to be said.

L327 Science not SCIENCE

Reviewer 2 Report

Thank you for this interesting article. I’ve requested changes which should significantly strengthen the paper. I hope this will help this paper realise its potential.

Abstract

needs expansion to signpost the main strands of argument, and final conclusions

Simple summary (a journal requirement)

missing

Intro

The expectation from the intro is that the article will focus on examining the merits of xenotransplantation, from the 3 perspectives of public health hazard, animal welfare, and the ethics of this form of biotechnology. However the focus of the final pages seems to significantly shift, into exploring public (particularly, American public) ignorance of science generally, the ignorance of scientists and the public re ethics esp. animal ethics, and the damaging effects of these on social consideration of biotechnologies such as xenotransplantation. This is a significant shift from the reader expectations set earlier, and focus of article until that point.

This needs to be resolved. The material not directly and clearly relevant to the specific issue of xenotransplantation could be removed from final pages. That would be a major change.

The other option is to correct the initial signposting of the article focus and structure, set by the abstract and Intro, to ensure the reader is aware the article will explore xenotransplantation initially, and then consider it as a case study to exemplify wider social concerns about science and ethics education, and the damaging impacts that deficiencies in these areas cause for social consideration of biotechnologies such as xenotransplantation.

  1. 1

The framework being applied seems to be encapsulated in ‘disambiguated three different aspects …  that I called “There are certain things humans were not meant to do,” “Rampaging monsters,” and “The plight of the creature.”’

It seems these three considerations are applied to xenotransplantation. To help the reader clarify the framework, illustrate this with a Figure – could Venn diagrams be used?

L 76 repetition of ‘epidemic’

L 92 ‘Another ethical issue associated with xenotransplantation is the question of how such organs would be distributed.’ Follow with something like ‘Inequitable distribution of medical resources within and between societies is a significant social problem [citations]’.

L 106 – 110

The ‘death is not a welfare concern’ view is outdated and significantly flawed. Must include here some brief discussion and acknowledgement that lack of awareness of future opportunity, does not mean a being is not disadvantaged, when deprived of that opportunity. This applies both to a child forced into child labour, rather than educated, and to an animal (or human) unknowing it its fate, who is then killed. And further, that death prevents the satisfaction of all interests/preferences, and also the experience of any positive future welfare states. Welfare is no longer considered just the avoidance of negative states. The opportunity to experience positive states is also now considered essential, to modern conceptualisations of good welfare (provide citation – e.g. Mellor). Death permanently eliminates all such possibilities. Accordingly, authors such as Yeates ‘death is a welfare issue’ (provide citation) have increasingly argued that far from not being an animal welfare concern, animal killing – even when painlessly idealised – is in fact a profound animal welfare concern.

L 146 for unfamiliar readers, clarify that pegs into holes analogises placing animals in confinement systems to which they are/are not naturally suited

L 176 – 180

Animal welfare is the state of welfare of an animal in a particular setting/circumstance. Scientific assessment of various parameters allow determination of this welfare state. Thus informed, one may consider what our practice should be. Different ethical theories may be brought to bear, but good welfare is prima facie desirable (i.e. w/o considering any wider contextual factors), under virtually all major theories. Having considered both welfare state and animal ethics, thus informed one may recommend change (or not) in social laws/policies/practices. (Hence, internationally, the vet specialisation is animal welfare science, ethics, law and policy - AWSEL).

i.e. the ethical considerations are distinct from the evaluation of animal welfare – from the welfare state of the animal.  

It is true that consideration of an animal welfare issue – i.e. a social issue – involves consideration of all 3 – welfare state + animal ethics + animal law/policy. And true that attempts to remove ethical considerations from discussions of animal welfare issues are both common, and very wrong. However, the distinctions between the above need to be clarified. The current text (esp. the final sentence ‘When we ask about an animal’s welfare…’) fails to elucidate these differences, adequately.

L 207

The swine are not ‘giving their lives for humans’. Their lives are being ‘taken’ for humans. Pls correct this.

L 208

‘… make sure that what life they do have is a good and pleasant one’ – provide some very brief suggestions as to how, e.g. adequate space, environmental enrichment, preservation as much as possible of social groupings, gentle handling by stockpeople with appropriate attitudes, training and licencing, prompt and sufficient vet care when needed, diet and climate control, etc. It need not be much more, or anything more, than this sentence.

L 289 – 298

Recheck – seems to be some confusion between facts and ethics. Ensure all correct as intended, and also as clear as possible

L 284 – which year?

L 308 - Typo

very minor formatting corrections needed throughout – spacing sometimes incorrect

Citations (even if self-citations) are needed for:

L 59 ‘As of mid-April 2019, an Israeli scientist has announced achieving three-dimensional printing of a human heart.’

L 70 ‘In 2016, NIH researchers announced that a pig heart had been kept alive in a baboon for three years.’

L 104 ‘the number of pigs killed in a year in the world for food is 769 million!’ – also state in which year

L 113 – 118 multiple citations needed to support the various claims

L 123 Aristotle and telos

L 128 husbandry

L 138 psalm

L 145 Hobbes

L 162 numerous accounts of AW

L 164 ‘my own account’ (self-citation needed)

L 166 Aristotle

L 185 – 189 FAWC definitions incl. 5 freedoms

L 215 Nature citation – cite and reference as normal

L 253 Newton

L 259 – 260

L 287-288

L 320

L 263 the Pope

L 398 ‘I have argued…’ (self-citation needed)
